# Intermolecular Interactions in the Formation of Polysaccharide-Gelatin Complexes: A Spectroscopic Study

**DOI:** 10.3390/polym14142777

**Published:** 2022-07-07

**Authors:** Svetlana R. Derkach, Nikolay G. Voron’ko, Yulia A. Kuchina

**Affiliations:** Department of Chemistry, Murmansk State Technical University, Murmansk 183010, Russia; voronkong@mstu.edu.ru (N.G.V.); kuchinayua@mstu.edu.ru (Y.A.K.)

**Keywords:** gelatin, polysaccharide, polyelectrolyte complexes, UV spectroscopy, FT-IR spectroscopy, ^1^H NMR spectroscopy

## Abstract

Gelatin, due to its gelling and stabilizing properties, is one of the widely used biopolymers in biotechnology, medicine, pharmaceuticals, and the food industry. One way to modify the characteristics of gelatin is molecular modification by forming non-covalent polyelectrolyte complexes with polysaccharides based on the self-organization of supramolecular structures. This review summarizes recent advances in the study of various types and the role of intermolecular interactions in the formation of polysaccharide-gelatin complexes, and conformational changes in gelatin, with the main focus on data obtained by spectroscopic methods: UV, FT-IR, and ^1^H NMR spectroscopy. In the discussion, the main focus is on the complexing polysaccharides of marine origin-sodium alginate, κ-carrageenan, and chitosan. The prospects for creating polysaccharide-gelatin complexes with desired physicochemical properties are outlined.

## 1. Introduction

The protein-polysaccharide polyelectrolyte complexes that are formed during biopolymer interaction are considered to be the basis for the creation of new materials in biotechnology, medicine, pharmaceuticals, the food industry, and other industries related to human health and nutrition [1,2,3]. The creation of materials is based on the self-organization principles of the complexes in the bulk of the aqueous phase [4]. The combined use of proteins and polysaccharides in the composition of the complexes contributes to the mutual enhancement of their physicochemical properties: their stabilizing ability and resistance to changes in external factors (pH, the presence of salts, changes in temperature) [5].

Nowadays, proteins and polysaccharides are widely used in the food industry [3,6] to encapsulate bioactive ingredients in functional food products [7,8]. These biomacromolecules as well as their complexes are used for protection and as delivery systems for bioactive food additives that can control their release and regulate their bioavailability [9,10].

Gelatin, a degradation product of the fibrillar protein collagen, is one of the biopolymers that is widely used in the food industry [11,12,13]. Gelatin has a unique thermoreversible gelation ability, which is accompanied by the macromolecule undergoing a conformational transition: coil↔collagen-like helix [14,15]. This property, along with the ability to interact with polysaccharides, is widely used to create various food systems based on polysaccharide-gelatin complexes that have the desired physicochemical properties [16,17]. Recently, new information on the structure and properties of various polysaccharide-gelatin systems (hydrogels [18,19,20,21] and composites [16], colloidal particles [22], emulsions [23,24,25,26], films [27], etc.) has come to light, expanding their range of applications in food technologies and products.

The interactions that take place between gelatin and polysaccharides play an important role in the development of new food systems. It is known that intermolecular interactions can lead to the formation of noncovalent polysaccharide-gelatin complexes, incompatibility, or covalent complexes [9,28]. Soluble (or insoluble) non-covalent complexes can be formed upon the physical mixing of gelatin and polysaccharide solutions, which is largely determined by the chemical nature of the biopolymers, pH, ionic strength, and mass ratio of biopolymers [29,30].

The driving force behind the formation of non-covalent complexes is the intermolecular interactions that take place between gelatin and polysaccharides, including electrostatic interactions, hydrogen bonds, and hydrophobic interactions. As a rule, electrostatic interactions between the oppositely charged functional groups of biopolymers play a decisive role [1]. Obviously, pH is the most important factor in the formation of soluble or insoluble polysaccharide-protein complexes [6,31]. The composition and properties of non-covalent complexes also depend on the polysaccharide/protein mass ratio and ionic strength since salts can weaken the electrostatic interactions of macromolecules [32].

This review analyzes the contribution of various polysaccharide-gelatin intermolecular interactions in the bulk of the aqueous phase during the formation of a non-covalent polyelectrolyte complex; attention is paid to the influence of polysaccharides on the secondary structure of gelatin. Polysaccharides from marine hydrobionts (algae and crustaceans) are mainly considered. The results of studies obtained by UV, FTIR, and ^1^H NMR spectroscopy are considered. Spectroscopic methods are among the most informative in the study of biopolymer systems since they not only allow for individual biopolymers to be identified and their structures to be analyzed, but they also provide information on the participation of individual segments of macromolecular chains in intermolecular interactions [33,34,35].

## 2. Properties of Gelatin and Polysaccharides

Because it is a polyampholyte, gelatin (G) is able to interact with both negatively and positively charged polyelectrolytes, forming non-covalent polyelectrolyte complexes in the aqueous phase. Fundamental studies on the formation conditions and physicochemical properties of complexes from synthetic [36,37,38,39] and natural polyelectrolytes [1,4,40], including polysaccharides [1,9,10], have been the subject of many works. As a rule, most attention is paid to the electrostatic interactions between oppositely charged groups of macromolecules, and to a lesser extent, to hydrogen bonds and hydrophobic interactions, which also need to be taken into account when forming biopolymer complexes. Gelatin consists of linear polypeptide built from series of up to 18 different α-amino acids [41,42], the names and structural formulas of which, as well as the amount per α-chain, are provided in Table 1. Glycine, proline, and hydroxyproline are the most abundant. Amino acids such as alanine, valine, isoleucine, leucine, phenylalanine, and proline, which are all a part of gelatin, contain hydrophobic radicals. The composition of gelatin includes polar groups: acidic (glutamic, aspartic acids) and basic (lysine, arginine, tyrosine, histidine). Amino acids such as hydroxyproline, hydroxylysine, and serine are capable of dissociating hydroxyl groups. Cysteine and tryptophan are not found in gelatin.

The presence of positively and negatively charged groups provides the possibility of electrostatic interaction with ionic polysaccharides. In this case, the positive groups are concentrated at the ends of the gelatin chains, and the negatively charged groups are distributed relatively evenly [14,41], which may be the reason for the different availabilities of polar groups upon complex formation with anionic and cationic polysaccharides [44,45,46]. Table 2 shows the experimental methods that are used to study the complex formation of gelatin with polysaccharides.

Among the polysaccharides that form complexes with gelatin, a significant place is occupied by polysaccharides from marine hydrobionts, which can increase the nutritional and biological value of food products. Below is a description of some of these polysaccharides.

κ-Carrageenan (C) is obtained from red seaweed (*Rhodophyta*) [75]. This high molecular weight polysaccharide exhibits the properties of a gelling agent and is characterized by antiviral properties [76]. The polysaccharide molecules consist of repeating carrabiose units—alternating blocks of 3-O-substituted 4-sulfo-β-D-galactopyranose (G4S) and 4-O-substituted 3,6-anhydro-α-D-galactopyranose (DA), sulfonated and nonsulfonated (Figure 1a). The blocks of the carrabios unit are connected by alternating (beta-1-4 and alpha-1-3) glycosidic bonds [77,78].

Alginates (SA) are obtained mainly from brown algae (*Phaeophyceae*) [79], including fucus algae (*Fucaceae*) [80,81]. Sodium alginate binds the atoms of heavy metals and radionuclides in the gastrointestinal tract and removes them from the human body [82]. Alginate molecules contain carboxyl groups and are linear binary copolymers that consist of 4-O-substituted β-D-mannuronate (M) and 4-O-substituted α-L-guluronate (G) residues linked by (1-4) glycosidic bonds [79,83] (Figure 1b).

Chitosan (Ch) is obtained by deacetylation from the natural polysaccharide chitin, the main component of the cuticle of arthropods (*Arthropoda*) [84,85]. Linear chains of chitin consist of chitobiose units linked by (β-1-4) glycosidic bonds (2-deoxy-2-acetamide-β-D-glucan residues). The chitosan molecule contains free amino groups in the composition of monomer units—deacetylated chitobiose units (residues of 2-deoxy-2-amino-β-D-glucan [85] (Figure 1c). The biological activity and the properties of the thickener and stabilizer determine widespread use of chitosan in the production of food and pharmaceutical products [86,87].

The intermolecular interactions of biopolymers (electrostatic interactions, hydrogen bonds, and hydrophobic interactions) leading to the formation of non-covalent polyelectrolyte polysaccharide-gelatin complexes can be studied using various methods (Table 2), among which spectroscopic methods occupy a significant place.

## 3. The Role of Electrostatic Interactions, Hydrogen Bonds and Hydrophobic Interactions

Analyses of the UV absorption spectra of gelatin solutions (Figure 2) show the presence of a wide absorption band and a wavelength corresponding to the absorption maximum, λ_max_ = 233–234 nm. The chromophore carboxyl groups of the *Asp* and *Glu* residues, hydroxyl groups, and unshared electron pairs of nitrogen conjugated with double bonds in the *His* and *Arg* residues contribute significantly to the position of the band [33,34,88], as do the conjugated double bonds in the benzene rings of *Tyr* [89,90]. The wavelengths corresponding to the absorption maximum in polysaccharide solutions lie in the far UV region compared to gelatin. For example, for κ-carrageenan, λ_max_ < 200 nm [68]; for sodium alginate, λ_max_ = 213 nm [52]; and for chitosan, λ_max_ = 224 nm [91]. This is explained by the presence of chromophores that are absorbed in the UV region of the spectrum: hydroxo-groups of polysaccharides, sulfo groups of κ-carrageenan [64], carboxyl groups of alginates, and amino groups of chitosan.

When a polysaccharide is added into a gelatin solution, a bathochromic shift λ_max_ from 233 to 237–240 nm occurs in the gelatin and is accompanied by an increase in the optical density and a significant broadening of the absorption band (Figure 2). The bathochromic shift in the gelatin spectrum can be explained by the biopolymer interactions at the molecular level with the formation of polyelectrolyte complexes, namely the electrostatic interactions of the sulfo groups of κ-carrageenan and the carboxyl groups of alginates with the basic *His* and *Arg* groups of gelatin, respectively. Protonated amino groups of chitosan interact with negatively charged carboxyl groups *Glu* and *Asp* of gelatin. Changes in the UV spectra were observed in the gum arabic-gelatin [61], and agar-gelatin [69] systems, which is associated with the formation of polyelectrolyte complexes. Clear evidence of biopolymer interaction is a pronounced general rise in the structureless absorption band (see Figure 2, right parts of spectra 5–7), which can reasonably be attributed to light scattering on the relatively large particles [52] of polysaccharide-gelatin polyelectrolyte complexes.

The ^1^H NMR spectroscopy method was used by [68,92] to study the structure of the complex formed by the anionic polysaccharide κ-carrageenan and gelatin as well as the mobility of individual functional groups. The one-dimensional nuclear magnetic resonance on the ^1^H nuclei (^1^H NMR) high-resolution spectra [33,34,35] is characterized by the high sensitivity of the chemical shifts to structural details. Figure 3 shows the ^1^H NMR spectra for a native κ-carrageenan solution, a native gelatin solution, and κ-carrageenan-gelatin aqueous mixtures of different κ-carrageenan/gelatin *w*/*w* ratios. The identification of ^1^H NMR lines was performed at the temperature (40 °C) when macromolecules of κ-carrageenan [75,93] and gelatins [14,94] are a statistical coil, and the side chains of the macromolecules are surrounded by solvent molecules.

Lines in the ^1^H NMR spectra of the polysaccharide (Figure 3) are assigned to the protons in the the carrabious block of κ-carrageenan according to the data in [95,96,97]. Lines in the ^1^H NMR spectra of gelatin are assigned to specific types of protons included in the certain amino acid residue in accordance with [35,98,99]. Spectrum analysis shows that the ratio of the integrated intensities of the individual lines of the spectrum is consistent with the content of amino acid residues in the gelatin macromolecule [100,101].

The study of the temperature effect on the integrated intensity of individual lines showed the following. For κ-carrageenan solutions, the integrated intensity of signals does not change with time at temperature 40 °C, and a similar picture is observed at 23 and 14 °C. For gelatin solutions, constant values of the integrated signal intensity are observed at 40 and 23 °C. At a temperature below the coil→helix conformational transition of the macromolecule, that is, at 14 °C, the intensity of the signals decreases, reaching a constant (equilibrium) value after temperature controlling the sample for three hours. At the same time, the noticeable (by about two times) decrease in signal intensity is observed for amino acid residues *Gly*, *Pro*, and *Hyp*. A similar result was noted in [100]. This is due to the fact that these amino acid residues play a significant role in the stabilization of triple collagen-like helices [11,102,103].

It is well known that similar change patterns can be observed in the ^1^H NMR spectra of aqueous solutions of gelatin over time at low temperatures (below the sol-gel transition temperature of ~20 °C) [98]. The coil→helix conformational transition of gelatin and the subsequent association of the helices to form a gel reduce the mobility of proton-containing groups in the helical fragments, leading to a significant decrease in the proton relaxation time. This leads to the absence of signals of such protons in the NMR spectrum. The gelation of gelatin is kinetic in nature since the macromolecular chains of gelatin in an aqueous solution have much greater mobility and flexibility in contrast to the rigid chains of κ-carrageenan [14,104]. The in-solution formation of a double helix in κ-carrageenan with subsequent gelation (when the temperature drops below 40 °C) occurs quickly and can be considered a first-order phase transition [93,105,106].

Analyses of the high-resolution ^1^H NMR spectra for κ-carrageenan-gelatin mixtures at different biopolymer weight ratios (*Z*, g_C_/g_G_) show the following: The proton signals of the *Val*, *Leu*, and *Ile* amino acid residues of gelatin are shifted ~0.01 ppm downfield at 40 °C (see Figure 3). Such a change in the chemical shift (δ) of the amino acid protons with developed hydrocarbon radicals (see Table 1) indicates a change in the hydrophobic interactions that involve these groups. At 14 °C, no such shift is observed; however, there is a downfield shift in the *Lys* proton signals by ~0.01 ppm and a high field shift of ~0.03 ppm for the *Hyp* proton signals [68]. A change in the chemical shift in the spectrum for the protons of the *Lys* amino acid residue containing a charged NH^3+^ group indicates a change in electrostatic interactions; and a change in the chemical shift for the protons of the *Hyp* amino acid residue containing a hydroxyl group indicates the breaking/formation of hydrogen bonds involving this group.

Figure 4 shows the equilibrium integral intensity (*I*) of proton signals of gelatin at the various mass *w*/*w* ratio (*Z*) of biopolymers and at different temperatures. A decrease in the integral intensity indicates the formation of rigid (slowly mobile) regions that do not contribute to the NMR spectrum [100], which indicates the intermolecular interaction of the biopolymer chains. A significant drop in the intensity of the *Val*, *Leu*, and *Ile*, *Hyp*, and *Lys* signals is observed at 14 °C.

Another important characteristic is the spin-spin relaxation time *T*_2_ (Figure 5) of the gelatin amino acid protons that are capable of forming various types of bonds when complexed with κ-carrageenan: electrostatic interactions (*Arg*, *Lys*, *His*), hydrogen bonds (*Hyp*, *Glu*), hydrophobic interactions (*Val*, *Leu*, *Ile*). *Pro*, which takes part in the formation of the triads *Gly–Pro–Y* and *Gly–Pro–Hyp*, is also of interest. Figure 5 shows the effect of carrageenan concentration (mass ratio of carrageenan to gelatin, *Z*, g_C_/g_G_) on the spin-spin relaxation time of gelatin protons at different temperatures. At 40 °C, in the low-value range of the mass ratio of the biopolymers (*Z* = 0.03 g_C_/g_G_), there is a sharp drop in *T*_2_ for all of the considered amino acid residues. Then, as *Z* increases up to 0.75, *T*_2_ remains practically unchanged [92]. 

At 23 °C, an increase in the mass ratio of the biopolymers to *Z* = 0.75 g_C_/g_G_ leads to a gradual decrease in *T*_2_ for all of the amino acid residues (Figure 5). Finally, at 14 °C, an increase in the κ-carrageenan content in the aqueous mixture causes a decrease in *T*_2_ for all of the amino acid residues in gelatin, with the exception of *Val, Leu*, and *Ile*. It should be noted that at 23 and 14 °C (but not at 40 °C), there is an especially strong decrease in *T*_2_ for *Lys*, the basic groups of which are capable of electrostatic interactions with the sulfo groups of κ-carrageenan.

Analyses of the high-resolution ^1^H NMR spectra and spin-spin relaxation times *T*_2_ of gelatin protons at different temperatures [68,92] made it possible to determine the contribution of various types of intermolecular interactions: electrostatic interactions, hydrogen bonds, and hydrophobic interactions, in the formation of non-covalent complexes.

Under conditions where gelatin macromolecules are in the conformation of a statistical coil (at 40 °C), hydrophobic interactions are mainly realized; in particular, they are realized between the non-polar sites of κ-carrageenan (for example, the –CH_3_ radicals of the methylated groups of the carrabious unit) and the hydrocarbon radicals of the residues *Val*, *Leu*, and *Ile*. Under conditions in which the gelatin macromolecule is in a helix conformation (at 14 °C), strong hydrogen bonds and electrostatic interactions are established between κ-carrageenan and gelatin. As a result, the value of the hydrophobic interactions is reduced to a minimum. These data are confirmed by the results obtained by UV spectroscopy and Fourier transform IR spectroscopy.

Complexation with κ-carrageenan also leads to a change in the secondary structure of gelatin and decreases the proportion of the ordered structures in gelatin (triple collagen-like helices), which is expressed in a decrease in *T*_2_ *Hyp*, *Glu*, and *Pro* at 23 and 14 °C (Figure 5).

## 4. Changes in the Secondary Structure of Gelatin upon Complex Formation with Polysaccharides

Fourier transform infrared (FT-IR) spectroscopy is widely used [107] to study the interactions between gelatin with polysaccharides. This method is an informative one that makes it possible to characterize the structural changes in biopolymer macromolecules during complex formation.

A number of papers present the results of studies on gelatins carried out using FT-IR spectroscopy. The influence of the nature of the sources [108,109,110] from which gelatins are obtained on the characteristic absorption bands of Amide I, Amide II, and Amide III is shown for various types of gelatins. The conformational changes in gelatin macromolecules from the skin of animals that occur due to temperature change have been studied [111]. In recent years, a large number of publications devoted to the study of fish gelatins have appeared, which is especially important for countries with a developed fishing industry. For example, in [13,15,17,112], the physicochemical properties were investigated, and it was shown that gelatin obtained from the skin and bones of fish can serve as a good alternative to the animal gelatin derived from the skin of bulls and pigs.

FT-IR spectroscopy is used to study the interactions between gelatin and polysaccharides [56,66,113]. In a study on the polyelectrolyte complexes of κ-carrageenan with the fish gelatin (Type A) used for the microencapsulation of Neem seed oil [114], shifts in the main transmission bands of the gelatin amide groups were observed.

Attributing the absorption bands in the FT-IR spectra of biopolymers to vibrations in the bonds of the corresponding functional groups and structural units is justified: for gelatin, see [33,35,109,111]; for sodium alginate, see [35,115,116,117]; for chitosan, see [56,118,119]; and for κ-carrageenan, see [120]. The absorption bands of the characteristic groups of gelatin and polysaccharides are shown in Table 3.

Figure 6 shows FT-IR transmission spectra of sodium alginate-gelatin mixtures with biopolymers at different mass ratios *Z* and g_SA_/g_G_. An analysis of the FT-IR spectra shows a shift to the low-frequency region of the Amide A band of gelatin from 3401 to 3392 cm^−1^ under the influence of complexation with sodium alginate. At the same time, the Amide A band of sodium alginate is shifted relative to gelatin towards higher wavenumbers, with values up to 3447 cm^−1^. In the case of chitosan, the Amide A band shifts to 3439 cm^−1^ [88,91]. The data obtained by FT-IR spectroscopy in [49,51] indicated the interactions that existed between gelatin-sodium alginate.

The shift of the Amide A band in gelatin can be explained by the formation of hydrogen bonds with polysaccharide molecules. Another reason is the electrostatic interactions between the carboxyl groups of sodium alginate and the basic *Arg*, *Lys*, *Hyl*, and *His* groups in gelatin [52] or the interactions between the amino groups of chitosan and the carboxyl groups *Glu* and *Asp* in gelatin [91,120]. This explanation is in good agreement with the results obtained by UV spectroscopy (see Section 3). A similar effect was observed in FT-IR spectroscopy studies on systems made of gelatin with chitosan [53,56,119], κ-carrageenan [64,65,66], gum arabic [62], gellan [72], konjac glucomannan [74], agar [69], pectin [70], and sodium alginate [48,49]. The structure analysis [50] indicated that there is strong interaction between sodium alginate and gelatin molecules resulted from intermolecular hydrogen bonds and ionic interactions. 

The data obtained by differential scanning calorimetry and FT-IR spectroscopy in [54,55] indicated the interactions that existed between gelatin-chitosan. It was found [57] that the complex formed between chitosan and gelatin was mainly through a hydrogen bond, but the size of the structure was also affected by electrostatic repulsions. The local structure (correlation length) and the global structure (large inhomogeneous structure size) in the composite solutions were found to be highly correlated to each other. The microscopy and infrared spectroscopy showed [65] that gelatin and κ-carrageenan molecules could be clustered in helices by electrostatic interactions and hydrogen bonding. Zeta potential and FT-IR analyses (at the different concentrations of Ca^2+^) revealed pectin-gelatin complex formation [71] through the electrostatic interactions and van der Waals forces that resulted in significant changes in conformation of macromolecules and microstructure.

Complexation with sodium alginate leads to the Amide I band (1653 cm^−1^, see Table 3) shifting in the gelatin spectrum towards lower wave numbers, sometimes shifting down to 1645 cm^−1^ at *Z* = 0.8 g_SA_/g_G_ (Figure 6). This indicates that there are electrostatic interactions that take place between the carboxyl groups of sodium alginate (1616 cm^−1^, see Table 3) and the amide groups of gelatin, as well as hydrogen bonds between biopolymers. A similar shift in the characteristic band has been shown in composite films [116] and complex membranes [51] based on alginate-gelatin mixtures as well as in κ-carrageenan-gelatin [67] and gellan-gelatin [66] films.

More evidence of interactions between oppositely charged groups of sodium alginate and gelatin is the shift in the symmetric stretching vibration bands of the COO^−^ groups in the IR spectrum of sodium alginate (1418 cm^−1^, see Table 3) to the low-frequency region (up to 1408 cm^−1^ at *Z* = 0.1 g_SA_/g_G_) (Figure 6). A similar result is shown in [50].

Similar changes in the FT-IR spectra were also shown for chitosan-gelatin systems [91]. Complexation with chitosan leads to a shift in the stretching vibration band of the carboxyl groups of the gelatin amino acid residues *Glu* and *Asp* (1165 cm^−1^, see Table 3) to a lower frequency region: 1157–1154 cm^−1^. A similar result was demonstrated in [56]. The appearance of this shift indicates an electrostatic interaction between the charged COO^−^ groups of the *Glu* and *Asp* residues in gelatin and in the NH_3_^+^ in the chitobiose units.

Polysaccharide interactions lead to a shift in the Amide III band in the gelatin spectrum, from 1238 cm^−1^ in the high-frequency region up to 1242–1244 cm^−1^ in the case of sodium alginate (at *Z* = 0.1–0.8 g_SA_/g_G_) [52] and up to 1243–1247 cm^−1^ in the case of chitosan (at *Z* = 0.1–1.0 g_Ch_/g_G_) [88]. This is associated with a decrease in the hydration of the gelatin macromolecules [121] and a decrease in the intermolecular interaction between the gelatin chains inside a collagen-like triple helix [108,122]. In other words, this effect characterizes a change in the secondary structure of gelatin, i.e., a decrease in the share of the α-chain in the triple helix conformation and an increase in the share of the α-chain in the conformation of a random coil. Anionic polysaccharide κ-carrageenan affects the secondary structure of gelatin in a similar way, as shown by FT-IR [21,67] and ^1^H NMR spectroscopy [68,92].

The results obtained by FT-IR spectroscopy suggest the following mechanism to describe the effect of ionic polysaccharides on the secondary structure of gelatin. The formation of non-covalent polyelectrolyte polysaccharide-gelatin complexes results in the fixation of the gelatin’s amino acid residues on the polysaccharide macromolecular chain. This partially blocks the mobility of gelatin macromolecules. In addition, although the density of the negative charge on the alginate chains and the positive charge on the chitosan chains decreases due to complex formation with gelatin, in general, polysaccharide-gelatin complexes carry the same charge: negative in the case of alginate, due to the gelatin carboxyl groups *Asp* and *Glu*, and positive in the case of chitosan, thanks to the basic groups *Arg*, *Lys*, *Hyl*, and *His*. It should be noted that gelatin’s carboxyl groups are ionized in the pH range of 5.2–5.6, which exceeds the pI (4.7) of gelatin, while the basic groups are ionized in the pH range of 3.4–3.9, which lies below the pI. As a result, the mutual electrostatic repulsion of polyelectrolyte complexes prevents the conformational coil→helix transition and, accordingly, causes a decrease in the content of collagen-like gelatin triple helices.

The noted decrease in the degree of the gelatin helix upon complexation with polysaccharides resembles a similar phenomenon found upon the association of gelatin with ionic surfactants [123]. Similar to the case of anionic alginate and cationic chitosan, low-molecular-weight ionic surfactants shield the opposite charges of gelatin, which causes the electrostatic repulsion of similarly charged polypeptide macromolecules and, accordingly, a decrease in the proportion of collagen-like triple helices.

## 5. Conclusions

The driving force behind the formation of non-covalent polyelectrolyte polysaccharide-gelatin complexes is the intermolecular interactions that take place between biopolymers, including electrostatic interactions, hydrogen bonds, and hydrophobic interactions. An analysis of the results of studies obtained by various spectroscopic methods (UV, FT-IR, and ^1^H NMR spectroscopy) made it possible to draw a conclusion about the role of intermolecular interactions under various conditions. Thus, it has been shown that at temperatures above the point of the conformational helix→coil transition (~20 °C) in gelatins, hydrophobic interactions play the main role in complex formation.

At temperatures below the conformational transition temperature, the main driving force is the electrostatic interactions between the oppositely charged groups of the gelatin polyampholyte and the polysaccharide (anionic or cationic). In this case, hydrogen bonds also play a certain role. It has been shown that complex formation with polysaccharides causes a change in the secondary structure of gelatin macromolecules, which manifests itself as a decrease in the proportion of the α-chain regions in the conformation of a collagen-like helix. An explanation is provided for the mechanism of such an effect.

The study of the intermolecular interactions in biopolymer systems containing gelatin and polysaccharides and the formation processes of the supramolecular structures of the complexes in the volume of the aqueous phase are relevant from the point of view of characterizing the behavior of nanodispersed systems during the interaction of the components that are capable of complex formation. Determining the principles for the formation of polyelectrolyte complexes with an optimal composition is the key to creating various systems with the desired technological and physicochemical properties that are in demand in the food industry.

## Figures and Tables

**Figure 1 polymers-14-02777-f001:**
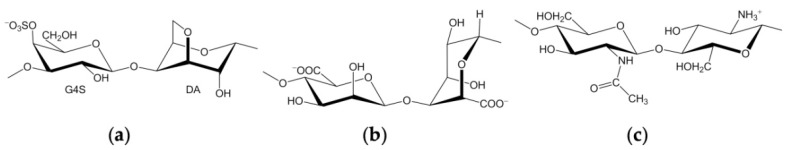
Structural formulas of polysaccharides from marine hydrobionts—(**a**) κ-carrageenan, (**b**) alginate, and (**c**) chitosan.

**Figure 2 polymers-14-02777-f002:**
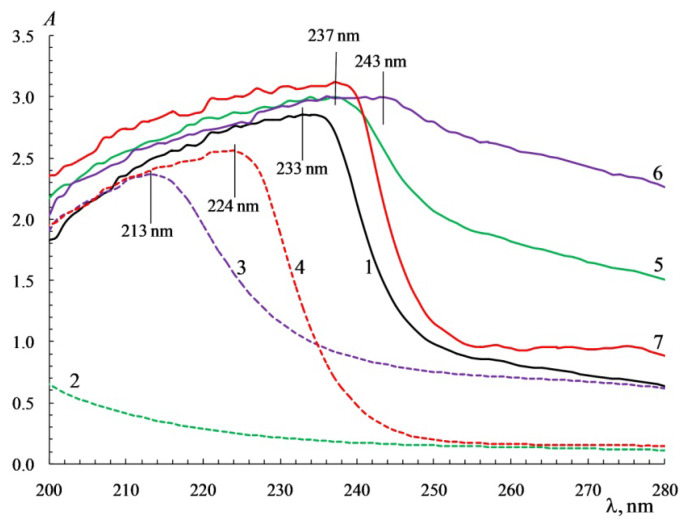
UV absorption spectra for native solutions of gelatin (*C*_G_ = 1.0 wt %) (1), κ-carrageenan (*C*_C_ = 0.5 wt %) (2), sodium alginate (*C*_SA_ = 0.5 wt %) (3), chitosan (*C*_Ch_ = 0.5 wt %) (4), and aqueous mixtures of κ-carrageenan-gelatin (5), sodium alginate-gelatin (6), and chitosan-gelatin (7). Polysaccharide/gelatin *w*/*w* ratio *Z* = 0.5 g polysaccharide/g gelatin, *C*_G_ = 1.0 wt %, 23 °C. Original figure.

**Figure 3 polymers-14-02777-f003:**
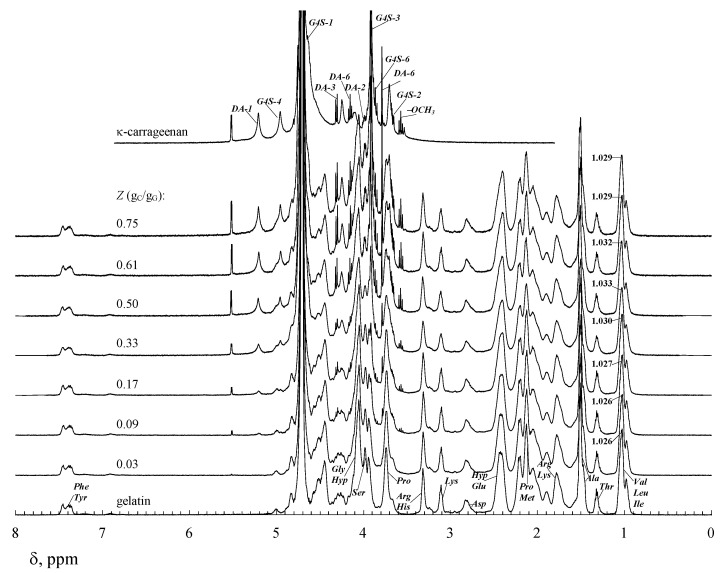
^1^H NMR spectra for native κ-carrageenan solution (*C*_C_ = 1.0 wt %), native gelatin solution (*C*_G_ = 2.0 wt %), and κ-carrageenan-gelatin aqueous mixtures of different κ-carrageenan/gelatin *w*/*w* ratios *Z*, g_C_/g_G_, in 99.8% D_2_O at 40 °C. Original figure.

**Figure 4 polymers-14-02777-f004:**
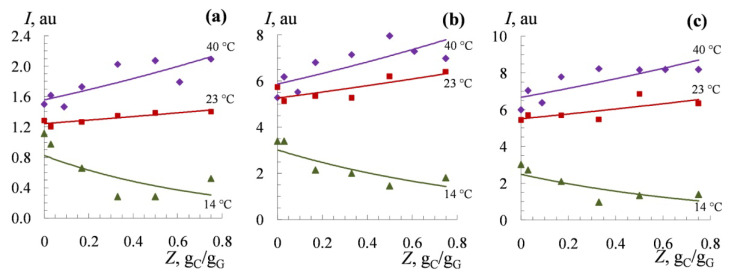
Integral signal intensity *I* for the protons of gelatin groups of *Lys* ε-CH_2_ (**a**); *Val*, *Leu*, and *Ile* γ-CH_3_ (**b**); and *Hyp* β-CH_2_ (**c**) at the κ-carrageenan/gelatin *w*/*w* ratio *Z*, g_C_/g_G_, at different temperatures. Solvent: 99.8% D_2_O; *I* represented in arbitrary units relative to the signal at 7.39 ppm at 40 °C. Original figure.

**Figure 5 polymers-14-02777-f005:**
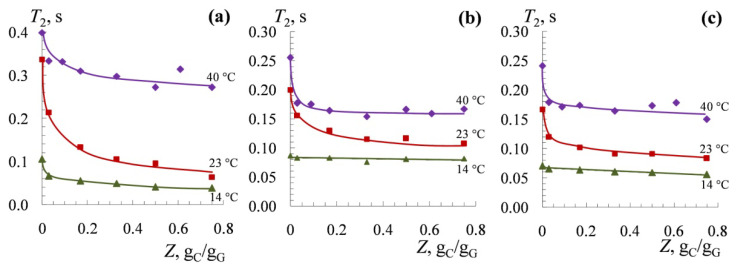
Spin-spin (transverse) relaxation times *T*_2_ for the protons of gelatin groups of *Lys* ε-CH_2_ (**a**); *Val*, *Leu*, and *Ile* γ-CH_3_ (**b**); and *Hyp* β-CH_2_ (**c**) in gelatin in 99.8% D_2_O at the κ-carrageenan/gelatin *w*/*w* ratio *Z*, g_C_/g_G_ and at different temperatures. Original figure.

**Figure 6 polymers-14-02777-f006:**
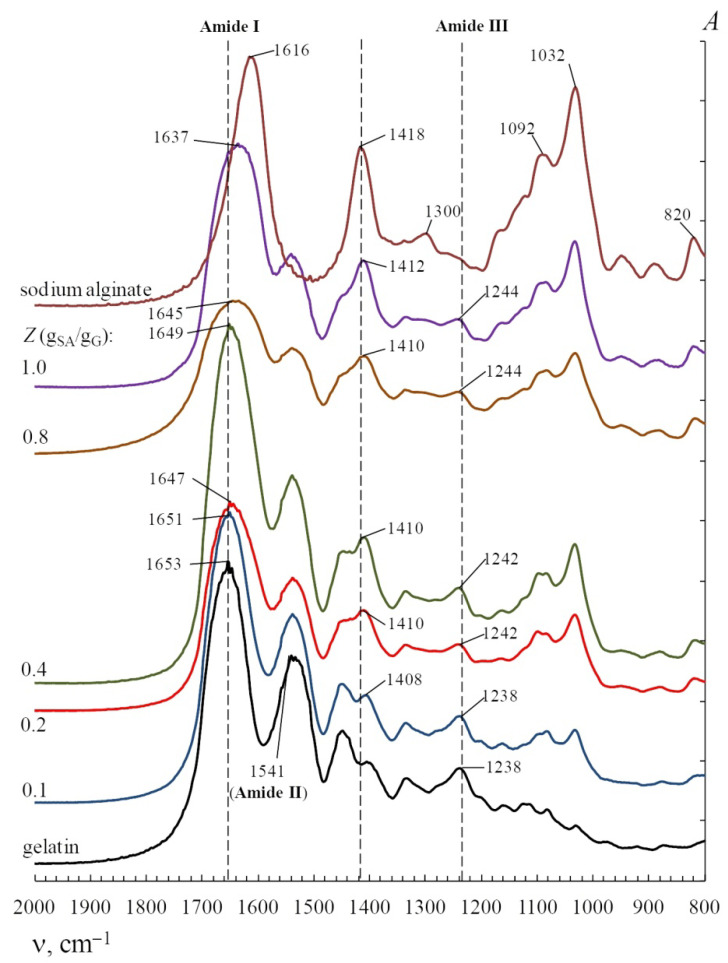
FT-IR spectra for native gelatin, native sodium alginate, and mixtures of gelatin and sodium alginate with different sodium alginate/gelatin *w*/*w* ratios *Z*, g_SA_/g_G_. Original figure.

**Table 1 polymers-14-02777-t001:** Amino acid composition of gelatin.

Amino Acid Residue	ChemicalDesignation	StructureFormula	Number of Amino Acid Residues per 1000
Mammalian Gelatin	Fish Gelatin
[41]	[14]	[15]	[43]
Glycine	*Gly*	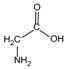	336.5	335	358	326
Lysine	*Lys*	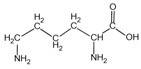	24.8	28	26	18
Hydroxylysine	*Hyl*	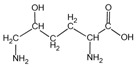	5.2	4	6	–
Histidine	*His*	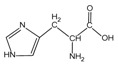	4.8	4	8	12
Arginine	*Arg*	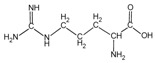	47.9	48	51	48
Aspartic acid	*Asp*	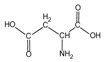	47.3	46	51	49
Glutamic acid	*Glu*	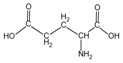	72.1	72	74	72
Serine	*Ser*	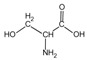	39.2	33	63	65
Threonine	*Thr*	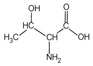	16.6	18	25	26
Hydroxyproline	*Hyp*	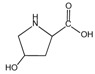	94.1	93	55	65
Tyrosine	*Tyr*	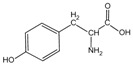	4.6	1	3	5
Alanine	*Ala*	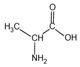	106.6	117	108	112
Valine	*Val*	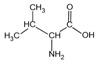	19.5	22	18	21
Leucine	*Leu*	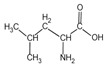	24.0	24	20	25
Isoleucine	*Ile*	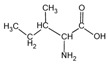	11.3	11	11	13
Proline	*Pro*	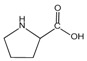	129.0	124	95	123
Phenylalanine	*Phe*	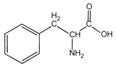	12.6	14	12	16
Methionine	*Met*	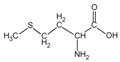	3.9	4	16	3

**Table 2 polymers-14-02777-t002:** Gelatin sources, polysaccharide types, and methods for study of polysaccharide-gelatin complex formation.

Gelatin Type and Sources	Polysaccharides	Study Methods	Ref.
Cold water fish skin (cod, pollock, and haddock)	Sodium alginate	Interfacial Tensiometry; Electrophoresis combined with Phase Analysis Light Scattering; Dynamic Light Scattering	[47]
Tilapia skin	Sodium alginate	Confocal Laser Scanning Microscopy; Atomic Force Microscopy; Dynamic Light Scattering; Phase Analysis Light Scattering; FT-IR Spectroscopy	[48]
Cold water fish skin (cod, pollock, and haddock)	Sodium alginate	Turbidimetric acid titration; Laser Doppler Electrophoresis combined with Phase Analysis Light Scattering; Dynamic Light Scattering; Confocal Scanning Laser Microscopy	[31]
Tilapia skin (260–270 Bloom)	Sodium alginate	FT-IR Spectroscopy	[49]
Bovine skin (M_w_ = 5 × 10^4^ Da)	Sodium alginate	FT-IR Spectroscopy; Wide-angle X-ray Diffraction; Scanning Electron Microscopy; Thermogravimetric Analysis; Differential Thermal Analysis	[50]
Porcine skin (Type A, 300 Bloom)	Sodium alginate	Scanning Electron Microscope; FT-IR spectroscopy; X-ray Diffraction; Differential Scanning Calorimetry; Positron Annihilation Lifetime Spectroscopy	[51]
Cold-water fish (pI 7.6, M_w_ = 13 × 10^4^ Da)	Sodium alginate	UV spectroscopy; Rheology	[43]
Bovine skin (Type B, 225 Bloom)	Sodium alginate	FT-IR spectroscopy; UV spectroscopy	[52]
Cold water fish skin (M_w_ = 6 × 10^4^ Da)	Chitosan (90% deacetylated)	FT-IR Spectroscopy; X-ray Diffraction; Scanning Electron Microscopy	[53]
Grass carp	Chitosan (95% deacetylated)	FT-IR Spectroscopy; Scanning Electron Microscopy	[54]
Bovine skin (Type B, 225 Bloom); Salmon skin	Chitosan	High Performance Liquid Chromatography; Differential Scanning Calorimetry	[44]
Cold water fish skin	Chitosan (75–85% deacetylated)	Differential Scanning Calorimetry; FT-IR Spectroscopy	[55]
Baltic cod skin	Chitosan (73% deacetylated)	Attenuated Total Reflectance Fourier Transformation Infrared (ATR FT-IR) Spectroscopy	[56]
Bovine skin (pI 4.9)	Chitosan (85% deacetylated)	Rheology; Small-angle Neutron Scattering	[57]
Fish skin (Type A, 240 Bloom)	β-chitin	FT-IR spectroscopy; Scanning Electron Microscopy	[58]
Cold water fish skin (Type B, pI 4.81)	Gum arabic	Rheology; Confocal Scanning Laser Microscopy	[23,59]
Bovine skin (Type A, 150 Bloom) Cold water fish skin (Type A)	Gum arabic; κ-Carrageenan	Electrophoresis; Rheology	[60]
Grass carp scales	Gum arabic	Intrinsic Fluorescence; UV-Visible Absorption Spectroscopy	[61]
Piramutaba skin	Gum arabic	High Performance Liquid Chromatograph; FT-IR Spectroscopy; Gel Electrophoresis SDS-PAGE; Scanning Electron Microscopy	[62]
Cold water fish skin	Gum arabic	Laser Doppler Electrophoresis combined with Phase Analysis Light Scattering; Turbidity; Dynamic Light Scattering	[63]
Bovine skin (Type B, 225 Bloom)	κ-Carrageenan	Turbidimetric Titration	[25]
Bovine skin (Type B, 225 Bloom)	κ-Carrageenan	ATR-FTIR Spectroscopy; Rheology	[21]
Pig skin (Type B)	κ-Carrageenan	Turbidity; Differential Scanning Calorimetry; Confocal Scanning Laser Microscopy; Phase Analysis Light Scattering	[29,30]
Tilapia skin (180 Bloom)	κ-Carrageenan	UV Spectroscopy; Dynamic Light Scattering; Atomic Force Microscopy; Confocal Laser Scanning Microscopy; FT-IR Spectroscopy	[64]
Bovine skin (240 Bloom)	κ-Carrageenan; Konjac glucomannan	Scanning Electron Microscopy; X-ray Diffraction; FTIR Spectroscopy; Rheology; Differential Scanning Calorimetry; Texture Profile Analysis	[65]
Tilapia fish skin (200 Bloom)	κ-Carrageenan; Gellan	Scanning Electron Microscopy; FT-IR Spectroscopy; Differential Scanning Calorimetry	[66]
Bovine skin (Type B, 225 Bloom)	κ-Carrageenan	UV spectroscopy; Rheology; FT-IR Spectroscopy; ^1^H NMR Spectroscopy	[21,67,68]
Cold water fish skin	Agar	UV Spectroscopy; FT-IR Spectroscopy with Attenuated Total Reflection (FTIR-ATR); Atomic Force Microscopy; Scanning Electron Microscopy	[69]
Grey triggerfish skin	Pectin	FT-IR Spectroscopy; Differential Scanning Calorimetry; Scanning Electron Microscopy	[70]
Tilapia fish skin (pI 9.58, 260–270 Bloom)	Pectin (low-methoxyl)	Spectrophotometry; Rheology; FT-IR Spectroscopy; Scanning Electron Microscopy	[71]
Tilapia skin (180 Bloom)	Gellan (low acyl)	Dynamic Light Scattering; Phase Analysis Light Scattering; Confocal Laser Scanning Microscopy; Rheology; FT-IR Spectroscopy	[72]
Tilapia skin (240 Bloom)	Gellan (low acyl)	Scanning Electron Microscopy; Rheology	[73]
Tilapia scale, (250 Bloom)	Konjac glucomannan	FT-IR Spectroscopy; Scanning Electron Microscopy; Rheology	[74]

**Table 3 polymers-14-02777-t003:** Location and assignment of the peaks identified in the FT-IR spectra of biopolymers.

Wavenumber of Absorption Band, cm^−1^	Absorption Band	Band Assignment
Gelatin		
3401	Amide A	Stretching vibrations of N–H and O–H groups
1653	Amide I	Stretching vibrations of C=O and C–N groups
1541	Amide II	Deformation vibrations of N–H groups and stretching vibrations of C–N groups
1238	Amide III	Stretching vibrations of N–H and C–N groups
1165		Stretching vibrations of COOH groups of *Glu* and *Asp* in gelatin
Sodium alginate		
3447	Amide A	Stretching vibrations of O–H groups
1616		Asymmetric stretches of COOH groups
1418		Symmetric stretches of COOH groups
1300		Stretching vibrations of C=O groups
1092		Mannuronic units
1032		Guluronic units
820		α-Configuration of the guluronic units
κ-Carrageenan		
3420	Amide A	Stretching vibrations of O–H groups
1263		Vibration of ester sulfate groups
928		3,6-anhydro-α-D-galactopyranose units
848		4-sulfo-β-D-galactopyranose units
Chitosan		
3439	Amide A	Stretching vibrations of N–H and O–H groups
1653	Amide I	Stretching vibrations of N–H and C=O groups
1560	Amide II	Stretching vibrations of N–H, C–N and C–C groups
1408		Asymmetric and symmetric stretches of CH_2_ groups
1261	Amide III	Stretching vibrations of N–H and C–N groups
1074		Skeletal C–O groups
1025		Skeletal C–O groups
854		β-Glycosidic bonds

## Data Availability

Not applicable.

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
