# Peer review of "Intermolecular Interactions in the Formation of Polysaccharide-Gelatin Complexes: A Spectroscopic Study"

_polymers, 2022, doi:10.3390/polym14142777_

Round 1

Reviewer 1 Report

Please refer to attached file

Author Response

Please, find the point-by-point response to the reviewer’s comments in the attached file.

Reviewer 2 Report

The authors have reviewed and summarized the recent analyses of the intermolecular interaction of hybrid materials composed of polysaccharides and gelatin using spectroscopy.

Whereas there have been numerous topics on biopolymers such as polysaccharides and gelatins, the authors have nicely focused on the most important molecular relationship between polysaccharides and gelatins.

This manuscript for the review paper is well-organized and contains appropriate up-to-date hot topics. The reviewer thinks that this review should be informative for readers of “Polymers.”

From these considerations, the reviewer recommends to accepting for publication in “Polymers,” if the following issues are resolved.

1)      The results in Figures 2-7: Are they the results cited from other research works performed by other researchers? Or are they the original results conducted by the authors? There is no information concerning the results shown in the manuscript.

2)      IR-Fourier” (page 1, line 15) and “Fourier IR” (page 18, line 392): Does this word mean FT-IR? What does this mean?

Author Response

(The authors gave the same response as above.)
